# Design of Scalable IoT Architecture Based on AWS for Smart Livestock

**DOI:** 10.3390/ani11092697

**Published:** 2021-09-15

**Authors:** Kristina Dineva, Tatiana Atanasova

**Affiliations:** Department of Modelling and Optimization, Institute of Information and Communication Technologies, Bulgarian Academy of Sciences, 1113 Sofia, Bulgaria; tatiana.atanasova@iict.bas.bg

**Keywords:** cyber–physical systems, IoT, cloud computing, AWS architecture, scalability, smart livestock farm, monitoring, stress test, Agile methodology

## Abstract

**Simple Summary:**

Due to the growing number of connected IoT devices, the scalability capacity and available computing power of the existing architectural frameworks would be reached. This necessitates finding a solution that meets the growing demands. Cloud-based IoT is becoming an increasingly popular and desirable solution. This work presents a specially designed architecture based on Amazon Web Services (AWS) for monitoring livestock using cyber–physical systems (CPS) and Internet of things (IoT) equipment and a wide range of cloud native services. Used services in AWS cloud are described in detail and their tasks according to the application area are clarified. A stress test to prove the ability of the developed architecture for data processing was completed. Experimental results showed that the proposed architecture with the services provided by Amazon is fully capable of processing the required amount of data and allows the CPS/IoT infrastructure to use automated scaling mechanisms.

**Abstract:**

In the ecological future of the planet, intelligent agriculture relies on CPS and IoT to free up human resources and increase production efficiency. Due to the growing number of connected IoT devices, the maximum scalability capacity, and available computing power of the existing architectural frameworks will be reached. This necessitates finding a solution that meets the continuously growing demands in smart farming. Cloud-based IoT solutions are achieving increasingly high popularity. The aim of this study was to design a scalable cloud-based architecture for a smart livestock monitoring system following Agile methodology and featuring environmental monitoring, health, growth, behaviour, reproduction, emotional state, and stress levels of animals. The AWS services used, and their specific tasks related to the proposed architecture are explained in detail. A stress test was performed to prove the data ingesting and processing capability of the proposed architecture. Experimental results proved that the proposed architecture using AWS automated scaling mechanisms and IoT devices are fully capable of processing the growing amount of data, which in turn allow for meeting the required needs of the constantly expanding number of CPS systems.

## 1. Introduction

There is a steady trend in increasing the number of animals on a farm. This growth in number varies around the world depending on the agriculture type and social structure. In Australia, there are on average 279 cows per farm, while in New Zealand, they average 440 [1,2]. Farms in Europe that keep more than 1000 animals such as farms in Estonia [3] are also becoming more common. An analogous situation is observed in the United States, wherein Michigan, Ohio, Indiana, and other traditional dairy states in the East and Midwest are built farms with upwards of 1000–5000 animals [4].

It has been found that the increase in herd size is driven by economies of scale—the cost of production per unit decreases with increasing herd size [5]. As pointed out in [5], the association between herd size and health and welfare is complex and affected by many factors. Providing farmers with access to rich data sources can aid in improving animal health and welfare. However, to achieve these goals, farmers must adopt modern technologies with an increased level of integration of automation processes and software management.

The development of technologies forces the transition to Agriculture 5.0 [6], where through automation and the introduction of recent technological solutions, traditional farm practices are modified and improved. As there is a constant need for development and due to the increasing trend in the number of animals in a herd/farm, software architectures need to be easily scaled to be able to receive a lot of data simultaneously from numerous Internet of things (IoT) devices. On the other hand, the current trend of increasing the urban population at the expense of people in rural areas raises the problem of reducing human participation in animal husbandry and the inclusion of more technological solutions to ensure humane breeding and careful monitoring of animal welfare. The advanced technological solutions led to the creation of smart farm systems using the communication capabilities of distributed and interconnected computing devices. Systems, where physical objects are represented in the digital world and integrated with computation, storage, communication capabilities, and are connected to each other in a network, can be defined as cyber–physical systems (CPS) [7]. Internet of Things technologies serves to integrate cyber–physical systems and to be an interface between CPS and their users. IoT is based on the idea of establishing a permanent connection between the physical and digital worlds [8]. According to ISO/IEC JTC1 2015, both terms, IoT and CPS, are interchangeable [9]. Both CPS and the Internet of Things are used to free up human resources, increase modern production efficiency, and be of significant help to improve product quality in smart agriculture [10]. It uses CPS and IoT technologies to increase productivity in smart farms through modern means in a continuously sustainable way to achieve the best in terms of quality, quantity, and financial return as well as to ensure that the process is environmentally friendly [11].

Different approaches for the design of software architectures in the field of livestock have been proposed. In [6], the usage of remote sensing (RS) technology in agriculture is discussed. It was pointed out that “the RS technology generates huge sets of data that necessitate the incorporation of artificial intelligence (AI) and big data to extract useful products, thereby augmenting the adeptness and efficiency of agriculture to ensure its sustainability” [12]. The reference architecture proposed in [13] has several layers and provides a model for farm management information systems. However, “the complete versions of the feature diagrams as well the detailed implemented architecture designs have not been shown” [13]. The study [14] discusses Global Sensor Network (GSN) as a foundation for the management of the streaming sensor data. In the proposed architecture, the signals received from the IoT sensors are collected by a gateway located on the farm and sent to smart farm servers through a high-speed broadband network. 

It is expected that IoT technology can make a breakthrough in livestock management by connecting the biological information of livestock and environmental information obtained by IoT sensors to farmers who are in a remote location on their farms via the cloud [15,16]. Attention was paid to the design of systems with analytical intelligence capability and data to be present on the premises. Systems for monitoring animals are available to farmers. Some of the approaches that are used in the design process are “domain-driven” [13]. The idea behind these was to make software development easier by providing a model for building flexible and reusable applications [17]. However, building the required infrastructure for a complex multi-layer architecture is very time consuming and, in many cases, is considered as an anti-pattern [18] and it is wise to just use another architectural approach.

An additional concept is the shifting from computing with centralised servers to distributed ones. One of these distributed technical infrastructures is presented by microservice architecture (MSA). The developed microservice architecture of a distributed IoT system discussed in detail in [19] consists of a group of microservices that communicate with each other synchronously or asynchronously.

Almost no current livestock systems extensively use the monitoring and security updates of remote IoT devices, which is a vital part of the system. Data security and sovereignty is essential nowadays as devices are exposed to different threats [20]. These aspects are deeply covered and explained in [21]. However, there is still a major problem to be addressed, providing centralised IoT device management where all remote location IoT devices can be configured remotely, and security patches applied as soon as the need arises.

Due to the growing number of connected IoT devices, the multi-layer and MSA architecture’s scalability capacities and the available computing power limits are about to be reached. As many IoT devices are intended for use in the smart livestock system and will generate large volumes of heterogeneous data with high variety and roughness, this imposes the need to find a solution that meets the constantly growing requirements. There is a need to use cloud computing to ensure a reliable and secure infrastructure that supports automatic scaling of resources according to the system needs and centralised IoT device management. Cloud-based Internet of Things (CB-IoT) [22] is becoming an increasingly popular and desirable solution. 

Recently, an interesting project was developed using CB-IoT by Amazon Web Services (AWS) for cattle management [23]. Within this project, the Australian team proposed a “Ceres Tag Management System” that houses the data and metadata on each ear cattle tag. The ear tag connects via satellite for data transmission, enabling the traceability of that tag throughout each animal’s life cycle. The proposed architecture focuses on data collection and transmission. The analytical processing of these data is outsourced. However, this project demonstrates that the use of cloud computing in this monitoring system is essential for achieving high speed, accuracy, and security in the data processing.

The aim of this study was to design a scalable cloud-based architecture for a smart livestock monitoring system following Agile methodology and featuring environmental monitoring, health, growth, behaviour, reproduction, emotional state, and stress levels of animals. The proposed architecture is capable of processing the required amount of data and allows the CPS-IoT infrastructure to use automated scaling mechanisms.

The rest of this paper is structured as follows. Section 2 introduces, step-by-step, the methodology that is followed and through which key features and requirements are identified to build a well-designed pure cloud architecture. In Section 3, the proposed AWS cloud-based architecture with its services and components is described in detail, focusing on scalability, availability, durability, and resilience. The section concludes with the architecture data ingestion pipeline and in-depth throughput load test with a demonstration and explanation of the achieved results. Section 4 presents a detailed discussion about the proposed architecture. Finally, Section 5 concludes the article.

## 2. Materials and Methods

The development of the proposed smart livestock monitoring system (SLMS) followed the established methodology—Agile (Scrum). Unlike other alternative methodologies, which have a linear and consistent approach aimed at completing a single project such as waterfall methodology, Agile methodology is an incremental and iterative approach, which separates a project into sprints, and helps complete many small projects [24]. Its application provides clarity about the sequence and cyclically of the steps of the complete process of building the system. This leads to a few benefits such as an accurate sequence of activities, good forecasting of deadlines for the start and end at each stage of the development process, clear knowledge of past, present, and future tasks, detection of potential difficulties at an early stage, smooth tracking of activities, and technologies used. The methodology also provides clarity about the methods, and algorithms for their implementation, determining the possibility of tasks for parallel or sequential implementation and providing information about the required number of people in a team [25].

Key stages of Agile (Figure 1) methodology are: 

Stage 1: *Requirements and analysis*—the requirements to which the monitoring system must meet are specified covers. This is an important stage in the entire process because it defines the macro framework of the system.

Stage 2: *Planning*—specify functional, non-functional, and technical requirements, infrastructure, programming languages, and others.

Stage 3: *Architecture design*—instructions for designing an architecture. This is an important stage in the complete process because it defines the skeleton of the system.

Stage 4: *Software development*—development of the individual parts (services) in the specified sequence.

Stage 5: *Architecture testing*—testing the durability, availability, reliability, and throughput of the system.

Stage 6: *Deployment* and monitoring system—verification and control of the process of implementation and subsequent renewal.

### 2.1. System Requirements

The developed system aims to provide a complete solution for smart livestock monitoring, which will allow a huge number of users to monitor their cattle. Therefore, the main system requirements are defined as:–Services—clearly distinguished services where each service performs a specific task without interfering with the tasks of any other services.–Maintenance—easy troubleshooting by isolating faults. This process does not interfere with the operation of the entire system but minimises human impact.–Flexibility—easily add or remove elements from the system without compromising its integrity.–Scalability—meet changes in demand by automatically creating or destroying IT infrastructure resources when there is observed increased or decreased traffic to and from the system. –Availability—minimum or zero downtime during infrastructure resources crashing or during prophylactic. The system must be available 24/7.–Reliability—ability to run and test the workload through its full lifecycle. –Agility—the system requirements must change quickly and inexpensively when needed.–Security—data and user accounts must be securely protected from unauthorised access.

### 2.2. Planning

A key point in the development of the architecture is the correct planning of the functionalities and the necessary technological stack for their implementation.

The functionalities supported by the smart livestock system are:

Functionality 1: Ability to integrate, maintain, and ingest data from a huge number of IoT devices from both newly created IoT systems and from existing ones regardless of the physical locations of the devices. The system aims to have the capacity to work with up to 10,000 IoT devices per second. 

Functionality 2: Possibility for distributed storage, analysis, and processing of big data—implementation of ETL processes with minimal latency and high resiliency.

Functionality 3: Ability for data modelling—build, train and deploy machine learning models in production. The aim is to create short-term and long-term forecasts for the health status of each animal based on the data collected by the IoT device placed near and on the animals.

Functionality 4: Ability for recognition of objects, people, and animals. Rapid spotting of a missing animal or an animal standing in a supine position for an unusually long period and other behavioural patterns, rapid identification in the event of unauthorised access by people or animals on the farm. 

Functionality 5: Possibility for analytics part, through which the collected data is visualised.

Functionality 6: Ensuring strong security of the entire system from unauthorised access.

Functionality 7: Ability to record the events that have occurred in the system (hardware and software), which show when and how they occurred and the results after their occurrence.

Functionality 8: Possibility of notifying farmers in the case of detection of abnormal deviation in the monitored parameters and animal behaviour.

### 2.3. Architecture Design

In the development process of the proposed cloud architecture, the five pillars of the well-designed architecture [26,27] were followed. They describe the key concepts, design principles, and the best architectural practices for designing and executing cloud workloads.

*Pillar 1: Operational Excellence*—The pillar of operational excellence focuses on the functioning and monitoring systems to deliver business value and continuous improvement of processes and procedures. Key topics specified in this pillar and implemented in the smart livestock architecture include automation of change (frequent, small, reversible changes), response to events, and setting standards for managing daily operations (performing operations as code).

*Pillar 2: Security*—The security pillar focuses on the protection of information and systems. Principals of this pillar are adopted in the smart livestock architecture for providing the tracking system confidentiality and ensuring data integrity, data segregation, and dedicated storage of sensitive data, management of what it can do with privilege management (create special roles for each farmer user), protecting smart livestock internal systems and establishing controls to detect various security events.

*Pillar 3: Reliability* —The pillar of reliability focuses on ensuring that the workload performs the intended function correctly and consistently when expected. The system uses several host zones in different regions, which ensures fast recovery from failures. The main topics from this pillar during the development of the architecture include the design of distributed systems, recovery planning, and how to deal with the change in demand. 

*Pillar 4: Performance Efficiency*—The efficiency pillar focuses on the efficient use of IT and computing resources. When building the architecture to cover the pillar, a pre-selection of the right types, number, and size of resources (AWS EC2 instances) is made based on the requirements for workload, performance monitoring, and making informed decisions to maintain efficiency as business needs evolve and increase.

*Pillar 5: Cost Optimisation*—The pillar of cost optimisation focuses on avoiding unnecessary expenses. In the process of developing the architecture of the system, preliminary calculations are performed, which requires an estimation of the type and number of resources as well as the cost analysis over time and scaling to meet the needs without overspending.

### 2.4. Software Development

Based on the described functionalities, the technological stack is determined, through which its implementation is achieved. The technology stack is the aggregation of all infrastructure and software tools and technologies used during the development process. Choosing the right technology stack is of great importance for obtaining successful overall results [28]. The following technologies are defined for use to meet the described functional requirements:

Infrastructure: To cover all system requirements presented in stage 1, the system was built entirely in AWS cloud infrastructure as this provider offers a well-developed open-source IoT edge runtime along with needed IoT cloud services [29]. They take care of the smooth and secure communication between IoT devices, and the services deployed in cloud infrastructure. 

Programming languages: The system performs tasks of diverse nature, and the use of different programming languages is recommended as each of them is better specialised in specific areas. The selected programming languages [30] are:–Python—high-level general-purpose programming language with an excellent SDK for developing cloud-based services [31]. It is used for the development of software deployed in IoT devices, and AWS Lambda functions, which are used as data processing triggers for AWS data pipelines.–HTML (Hypertext Markup Language)—a standard markup language for documents designed to be displayed in a web browser [32]. Used in the system to develop the body of UI HTML components.–CSS (Cascading Style Sheets)—style sheet language used for describing the presentation of a document [33]. Used for the visual part of the UI.–JavaScript—high-level, multi-paradigm, and object-orientation programming language [34]. It is used in the system both to create UI interactivity and to create specific Lambda functions in AWS.

Services: The smart livestock architecture built in AWS is a combination of standalone AWS serverless services and custom ones. Each of them performs an independent specific task. Certain groups of services communicate and exchange information with each other asynchronously [35]. The groups of services [36] required to perform the system functionalities are shown in Table 1.

### 2.5. Architecture Testing 

Before deployment in a production environment, it is necessary to test the whole architecture to isolate potential issues and problems. This is a highly critical step. It is important that enough tests are performed on all sensitive points to confirm the sustainability of the developed architecture, its proper functioning, and its safety for deployment in production [37]. 

#### 2.5.1. Scope

The scope of architecture testing covers the most critical data ingestion rates and performance metrics of the architectural prototype of smart livestock and includes:–System real-time throughput ingestion rates for data pipelines. –System storing rates for real-time data.–Serverless functions for performance metrics.–Total average latency of the data pipelines during continuously high loads.–Total count of errors of the data pipelines during continuously high loads.–Overall data pipeline durability and scalability and performance.

#### 2.5.2. Testing Strategies

Data load testing will ensure the system is ready for ingesting high volumes of data for a continuous-time period, and that average latency and error counts are within the expected limits.Data integrity testing will ensure that the data have been populated as intended to all locations and all insertion events occurred properly in the correct sequence.Testing on functions will ensure the proper data acceptance, processing, and retrieval within the system during continuous high loads.Stress testing will ensure the system’s durability, scalability, and performance.Security and access control testing will ensure that the ingested by the system data are accessible only for users with the right permissions.

#### 2.5.3. Results Evaluation 

Assessments of test completion criteria and success criteria by analysing test logs and various performance metric charts. 

### 2.6. Deployment

The architecture deployment stage was performed using Terraform. Hashicorp Terraform [38] was used for the creation, modification, and provisioning of the resources in AWS infrastructure needed for the smart livestock architecture. For that configuration, files in JSON format are used where resources and providers are defined. Then, certain commands are called, which change the current state of the infrastructure according to the needs. Robust Terraform capabilities can be used to restore the infrastructure state if required.

## 3. Results

### 3.1. Used Services in AWS Cloud

The proposed architecture designed for smart livestock farming is shown in Figure 2. It was built with AWS serverless services and contains several groups of services such as compute, storage, databases, analytics, network, mobile, management, developer, IoT, security, enterprise application, and others. The necessary services for creating a scalable and robust livestock monitoring system were grouped in frames. These were: AWS IoT Core, Lambdas, Data Recognition, Streaming Data, Data Lake, Data Stores, Data Processing, Machine Learning, Notifications, Analytics, Logging, and User Identities. The frames were developed to satisfy the required functionalities described in the Materials and Methods section as Stage 2. The designed architecture complies with all pillars of the well-designed architecture described in the Materials and Methods section on Stage 3.

#### 3.1.1. AWS IoT Core Frame

The AWS IoT core consists of five services that maintain the needs of all IoT devices, connect to AWS cloud, manage devices, update over-the-air (OTA) [39], and secure the IoT devices. It uses the TLS communication protocol to encrypt all communication [40]. Services in this framework are rules, topics, shadow service, AWS IoT device defender, and AWS IoT device management.

The Rules enable the IoT devices developed for smart livestock to interact with AWS services.

Some of the rules used in the system are:–Filtering data coming from IoT devices;–Separation and recording of data according to their type in various kinds of databases;–Sending notifications to users in certain circumstances (for example, occurrence of abnormal events in the monitoring process);–Real-time processing of messages coming from a large number of IoT devices from different locations;–Setting alarms to notify the user when reaching predefined limits of certain parameters (for example, reaching a critical battery level of IoT devices);–Send the data from a Message Queuing Telemetry Transport (MQTT) message to a machine learning frame to make predictions based on the ML model;–Send message data to an AWS IoT analytics channel; and–Send data to a web application.

AWS core default implementation is based on the MQTT protocol, which is used by AWS core to interact with devices. It decouples the producer and consumer by letting clients publish, and having the broker decide where to route and copy messages. Rules are analysed and actions are performed based on the MQTT topic stream. The Topics identify AWS IoT messages. A message broker is used to apply topic names and topic filters to route messages send using MQTT and Hypertext Transfer Protocol (HTTP) to the Hypertext Transfer Protocol Secure (HTTPS) message URL. First, it is important to create a hierarchy of topic names that identify the relevant IoT system. Topic names and topic filters are UTF-8 encoded strings. A topic name could refer to a sensor in developed IoT devices (for example, sensor/temperature/caw1/farm1). After forming hierarchical identifying topic names, topic filters are created, which filter the messages by topic names and send them to the services subscribed to them.

As many IoT deployments consist of hundreds or thousands to millions of devices, it is essential to indicate (shadow) IoT devices. AWS shadow service creates a shadow identification of each IoT device used in the smart livestock frame. In this way, each of the used devices can be accessed and managed specifically by different services. These changes are made via the MQTT protocol or via HTTP using the device shadow REST API [41] (p. 525).

Due to the specifics of the used IoT devices in this application, they have limited computation, memory, and storage capabilities. This restricts opportunities for security. Therefore, additional measures are required to ensure the security of the devices. AWS IoT device defender [36] (p. 31) was chosen in this application. Registering the IoT livestock device with an AWS IoT core provides the digital identity of the IoT livestock device in the cloud. The IoT livestock devices use the certificates to identify themselves to the AWS IoT core. These certificates are generated by the AWS IoT core. From this moment, the AWS IoT device is responsible for maintaining and implementing IoT configurations such as device identity, device authentication, device authorisation, and device data encryption. This is an out-of-the-box solution as this service provides the necessary protection for all devices by constantly checking for deviations of IoT configurations and the occurrence of abnormal deviations from the expected behaviour of the device. In the case of abnormal behaviour detection, the user and CloudWatch service are immediately informed.

AWS IoT device management is a service that remotely manages IoT devices (individually or in groups) when a signal is detected for abnormal behaviour in the devices. This service eliminates problems by managing and updating the software and firmware.

#### 3.1.2. Lambda Frame

A Lambda function is a stateless piece of code, with an input and output that can be triggered from a wide array of sources internal and external to AWS. It can be used to automatically scale an application up and down without making capacity planning decisions. Unlike an EC2 instance [42], a Lambda has one dedicated purpose and intentionally only runs for up to a few minutes. Lambda functions scale instantly to hundreds of instances, with almost no platform maintenance. 

In this work, Lambda functions in the livestock monitoring system are responsible for:–Converting each frame to jpeg format if it is different;–Storing raw frames in Amazon S3 by generating a unique key for each frame, which contains specific data about the place and time of the shooting: frames/farm_1/year/month/day/hour/min/sec/frame_number.jpeg;–Transmitting frames in jpeg format to Amazon “Rekognition” frame;–For each frame, Lambda functions saves to Amazon DynamoDB specific metadata such as a unique ID, S3 bucket and key, where the frame is saved, the approximate recording time and more.

#### 3.1.3. “Data Rekognition” Frame

“Data Rekognition” framework deals with data coming from the cameras in the smart livestock IoT system. Each recorded video is divided into separate frames so that they can be processed and analysed in near real-time.

AWS Rekognition Service analyses objects, people, animals, scenes, and activities contained in the collected videos. Through this service, the application performs counting and identification of the presence or absence of specific animals, people, or objects around the farm in real-time. Personal protective equipment (PPE) detection is also performed. The results of the performed analyses are sent for storage in NoSQL database service DynamoDB, from wherein the presence of certain abnormal phenomena (identification of a missing animal, abnormal behaviour of an animal, or the presence of a stranger on the farm) are sent to Amazon Pinpoint via Kinesis Published Events.

AWS Rekognition is configured to detect and recognize faces in streaming video and to detect and count livestock. It uses AWS Kinesis Video Streams to receive and process a video stream, which is H.264 encoded. Used events are face detection and face comparison, and livestock label detection.

AWS Rekognition is used currently for several cases, and different performances have been observed. It has high accuracy when identifying persons, counting animals entering, or leaving an area one by one. However, the accuracy drops significantly for animal detection, especially in the part where the animal head position (high/low) is of interest. The results are also influenced by the livestock density occupying the same areas.

#### 3.1.4. Notification Frame

Amazon Pinpoint [36] (p. 49), as a communication service, connects with users through various channels such as email, SMS, push, or voice. This service is used to personalise messages with the right content. It sends push notifications to the smart livestock application after pre-provided data that authorises PinPoint to send messages. The credentials that are provided depend on the operating system:–For iOS apps, an SSL certificate is provided. The certificate authorises the PinPoint service for sending messages to the smart livestock apps.–For Android apps, a web API key is provided. These credentials authorise the PinPoint service for sending messages to the smart livestock apps.

AWS AppSync takes care of managing and updating real-time data between web and mobile app and cloud. Additionally, it allows apps to interact with data on mobile devices when it is offline.

#### 3.1.5. Streaming Data Frame

The data coming from various groups of IoT devices, containing sensors for measuring various parameters of indoor and outdoor environments in the farms and monitoring certain parameters for the condition of each animal is sent to the Kinesis Data Streams Service (KDS). In it, the collected data are analysed for real-time anomalies and, depending on the result, sent to the Kinesis Data Firehouse Service (in the absence of detected anomalies), where the data are automatically scaled, grouped, compressed, transformed, and encrypted before being stored in the simple storage service (S3).

#### 3.1.6. Data Stores Frame

In case of detected abnormal behaviour, KDS sends the data to the data stores (operational and analytical), where purpose-built databases like DynamoDB, Redshift, and serverless architecture are used to store events, deliver microservices (API Gateway and Kinesis Published Events), and generate events for an operational data store. Some of the data are sent to a real-time operational dashboard leveraging microservices and AWS AppSync. Alerts are delivered to multiple channels using Amazon Pinpoint where it can grow and scale globally to billions of messages per day across channels.

#### 3.1.7. Data Processing Frame

The data stored in the data stores frame were sent for analysis and processing to the data processing frame. There is an ETL (Extract Transform Load) process for data integration, which refers to three main steps: extract, transform, and load [43]. This process is performed through the AWS glue service that defines and orchestrates the ETL workflow. Glue service uses the AWS Glue Data Catalog to store metadata about the data source and transforms. AWS identity and access management (IAM) policies [44] are used to control access to data source management. These policies allow separate groups of farmers to safely publish data to their vets while protecting sensitive information.

#### 3.1.8. Data Lake Frame

In order to extract more value from the collected data, it is necessary to store them both before their processing (raw data) and after their processing (performed transformation, normalisation, balancing, etc.). The exponential growth of data from various sources creates difficulties in storing (raw data and transformed data) and analysing big data. Therefore, the processed data are sent for storage in the data lake frame.

Data lake is used to store raw data and to create curated processed data in Amazon simple storage service (S3) using AWS Glue and Amazon EMR. Amazon EMR de-couples compute and storage, allowing both to grow independently, leading to better resource utilisation. EMR allows farmers to store data in S3 and run compute as needed to process the data. If necessary, clusters can be created. These clusters can automatically resize clusters to accommodate peaks and scale them down without impacting S3 data lake storage. Additionally, it is possible to run multiple clusters in parallel, allowing each of them to share the same dataset. With EMRs, it monitors cluster retry and failed tasks and automatically replaces poorly performing instances (EC2) using Amazon CloudWatch.

#### 3.1.9. Logging Frame

Amazon CloudWatch collects and tracks metrics, logs, and audits, sets alarms and automatically reacts to changes made in each AWS service, which is part of a smart livestock system. It can be used for triggering scaling operations or can also be used for providing into the deployed resources. AWS ElastiCache [36] (p. 11) provides metrics for monitor created clusters from the Amazon EMR service in the data processing frame and can be accessed through CloudWatch. ElastiCache provides both host-level metrics (e.g., CPU usage) and metrics that are specific to the cache engine software (e.g., the cache gets and cache misses). These metrics are measured and published for each cache node in 60 second intervals. Additionally, it improves application performance through caching the most heavily requested items, which provides low latency. Clusters can be automatically deleted upon task completion.

#### 3.1.10. Machine Learning Frame

Rarely used data are stored in S3 Glacier (Serverless) [36] (pp. 61–62), which is used for the long-term archiving of the data. It is not readily accessible as S3 Buckets. It should only be used for content that is to be archived. If necessary, these data are unzipped and returned to S3. Then, it can be successfully used in the machine learning frame, where the data train machine learning algorithms for regression or classification prediction. Amazon SageMaker [36] (pp. 33–34) was used to build, train, and deploy interface models. To predict the future health of the animals, the boosted decision tree algorithm was trained, which takes as parameters the animal’s temperature, heart rate, duration of activity and lying down, and others. With these data, the model is trained to make a binary prediction of one of two classes: the animal has good vital signs, and the animal has poor vital signs. In addition to the classification of the health status of the animals, a regression is performed to predict, in the short-term, the future amount of milk that will be extracted from each animal. Linear regression is the algorithm trained with the historical data stored in the S3 Bucket. The trained models are deployed, and their self-learning continues through new data coming from the data stores frame.

To better manage the battery power consumption of each IoT device, another training of the ML model was performed. This was undertaken to customise the frequency of use (inclusion of IoT devices and perform real-time monitoring) of IoT devices and provide a prediction of the battery charge level. These models are then deployed to the edge models on AWS IoT Greengrass core [36] (p. 29).

#### 3.1.11. Analytics Frame

To perform data analytics, Amazon Athena was used. It allows for the analysis of data stored in Amazon S3 Bucket using standard SQL statements. Athena is a query engine. The results are displayed in Amazon QuickSight [36] (p. 12), which is a scalable, serverless, embeddable, ML-powered business intelligence (BI) service. Additionally, it is a super-fast, parallel, in-memory, calculation engine. Real-time data from Redshift is also included in dashboards. Developed dashboards can be accessed by affiliated farmers regardless of their location. Dashboards contain personalised information about each animal, which is visualised in tabular form and diagrams. From dashboards, farmers have the opportunity to monitor the animal’s growth and their vital signs. An option to compare the growth by days, months, and years is also included.

Athena is used to gain analytical insights over raw data stored in AWS S3, but not in real-time. Such examples are analytics over device data and performance (e.g., find all devices with sensor readings above/below the threshold, find several faults/null sensor readings, aggregate comparable results for a certain period, etc.). For IoT devices, the performance and accuracy are highly important and by analysing the raw data, insights can be gained about that.

#### 3.1.12. Presentation Frame

The developed application is responsible and accessible from all devices. Amazon Route 53 (Domain Name System-DNS) [45] is used to make the application available to users. It effectively connects user requests to infrastructure running in AWS—Amazon Elastic Beanstalk, Elastic Load Balancer, and EC2 instances. AWS Elastic Beanstalk was used to deploy and scale our dynamic web application. Elastic Load Balancer (ELB) automatically distributes incoming application traffic across multiple EC2 Instances. In the smart livestock monitoring system, two reserved instances are used, but if more computing power is needed, additional spot instances are picked up, which are closed as soon as the calculations are completed. With the use of ELB application, greater levels of fault tolerance are achieved seamlessly, providing the required amount of load balancing capacity needed to distribute application traffic.

#### 3.1.13. User Identities Frame

It turns out that data acquired from various IoT sensors are valuable enough to have the power to change the overall business models and behaviour of entire industries. Therefore, the future of IoT systems depends on the security capabilities that systems can provide to their users. Users must go through Amazon Cognito User and Identity Pools to access the content, which provides security features such as multi-factor authentication (MFA), checks for compromised credentials, account takeover protection, and phone and email verifications. AWS services provide protection of the users’ data, accounts, and workloads from unauthorised access with encryption and key management. The threat can be detected through continuously monitoring and protecting accounts and workloads within the cloud environment by logs.

### 3.2. Smart Livestock IoT Devices Prototypes

#### 3.2.1. IoT Devices

Livestock remote monitoring is performed through a specially designed IoT devices shown on Figure 3. They consist of sensors, logical blocks, communication components, power supplies, and video surveillance (photo, thermal, and video cameras). The purpose of the system is to collect data from hundreds or thousands of IoT devices at defined intervals (e.g., 30 min) or in real-time.

In the system there are three types of participants:

#### 3.2.2. Livestock IoT Device

A set of sensor groups, logical blocks, and power supply collected sensor readings from livestock for temperature, humidity, barometric levels, gyroscope, noise, and GPS coordinates, which will be further extended to also collect readings for heart rate and gas analysis. The communication in each livestock IoT device is device-to-edge using Long Range (LoRa) or Transmission Control Protocol (TCP) based protocols. All devices on a farm have strict firewall rules. When performing initialisation, all incoming requests are deactivated. Incoming requests are then only allowed for those that come from the IoT Edge IP address on this farm. Outgoing requests are only allowed up to the IP address of the IoT Edge. Livestock IoT devices will not be able to communicate with each other on their farm or with other devices. In case the IoT device cannot connect to the IoT edge, or the data transmission is obstructed by any other reasons, the device can keep the sensor readings until the moment it reconnects. Then, the IoT device will send the up-to-date data. A prototype of the IoT device was especially designed and developed for testing purposes (Figure 3a). 

#### 3.2.3. Cameras—Photo, Thermal, and Video Cameras 

The cameras are used to monitor farm animals. There are two types of cameras included in the system: video cameras and thermal cameras. As monitoring quality both during the day and at night is an important indicator for assessing the efficiency of the video surveillance system, thermal imaging devices are a great advantage due to their ability to convert heat into an image visible to the human eye. They balance visible light with an infrared connection. This allows users to effectively monitor an area of the farm in all lighting conditions. The recordings are sent directly to the IoT Edge. Cameras are ready-made external components. Software can be installed that enables the communication between the cameras and the IoT Edge device.

#### 3.2.4. IoT Edge 

A set of a group of sensors, logical blocks, and power suppliers collected environmental sensor readings—air pressure, temperature, humidity, light, and human detection. It also collects data from livestock IoT devices and cameras. 

A prototype of the IoT Edge was especially designed and developed for testing purposes (Figure 3b). The AWS IoT Greengrass core was installed, which is an Internet of Things (IoT) open-source edge runtime and cloud service that helps to build, deploy, and manage device software [46]. AWS IoT Greengrass core was used to manage local processes, communicate, and synchronise certain groups of devices and exchange tokens between Edge and cloud, which acts as a hub or gateway in Edge. The communication is Edge-to-Cloud using TCP based protocols. It consists of MQTT Broker, Local Shadow Service, AWS Lambda, Meta Data, and Trained Models. Through AWS IoT Greengrass core, the following tasks are performed:–Processing of large data streams and automatic sending them to the cloud through the local implementation of Lambda functions (AWS Lambda);–MQTT messaging over the local network between livestock IoT devices, connectors, and Lambda functions using a broker (managed subscriptions). Additionally, MQTT messaging communication between livestock IoT devices and AWS IoT core (MQTT Broker);–Secure connections between livestock IoT devices and cloud using device authentication and authorisation (Meta Data);–Local Shadow synchronisation of devices (Local Shadow Service);–Deployment of cloud-trained machine learning models for regression that predicts a percentage of the future power of the battery in relation to the individual frequency and load of the monitoring livestock system (Trained Model);–Automatic IP address detection enables livestock IoT devices to discover the Edge device (Topics); and–Updated group configuration with secured over-the-air (OTA) software updates.

As an external terminal for the system, IoT Edge plays a key role as a mediator. All communications external to the system are controlled by it and determines the high level of security it must have. All external requests to IoT Edge that are also external to the system are encrypted using TLS/SSL. All outgoing requests that are external to the system must also be encrypted. Edge collects and stores the information provided to it by IoT devices and cameras until the information is sent to the cloud. Edge plays a significant role in reducing and transforming the information into the required type and volume, thus significantly reducing the amount of outgoing Internet traffic from the system. This is an important advantage in the reliable operation of remote IoT systems.

### 3.3. IoT Device Communication

The identification of these three types of participants was due to the Edge-enabled cloud computing platform to combine large-scale low-latency data processing for IoT solutions. Edge allows for the coordination and management of all resources involved in the livestock IoT system as well as the absorption of data from various sources in the cloud.

AWS Greengrass is used to extend functionalities by allowing devices to act locally on the data they generate while still taking advantage of the cloud. There are two types of IoT livestock devices. One type of device uses WiFi and communicates with the Greengrass Edge by pushing messages to the local MQTT broker. The Greengrass Edge device uses the MQTT component to transfer data to the AWS IoT core and back to the IoT livestock devices. In addition, the authors included local Lambda functions in the local communication between IoT devices and the Greengrass Edge.
IoT devices push messages to a local topic to which a local Lambda is subscribed. This lambda pushes the message to the AWS IoT core topic.AWS IoT core pushes a message to an IoT device using the topic. The local Lambda is subscribed to this topic and receives the message. The local Lambda then pushes the message to a topic to which the IoT device is subscribed.

This approach allows full control over the messages generated by IoT devices. As local Lambdas are deployed remotely, it becomes easy to make changes in the Lambdas logic that controls how IoT messages are handled.

### 3.4. Architecture Data Ingestion

A leading role of the smart livestock system architecture working with data pipelines is to sustainably gather and prepare information in forms that allow the other parts of the system to perform their tasks quickly and efficiently. As the data coming from distributed IoT systems (part of the smart livestock system) is in real-time and often extremely large and heterogeneous, the data pipeline scalability and performance capabilities are considered critical. For this reason, serverless cloud data pipelines were chosen to develop this system, because they can provide the scalability and resilience needed and significantly reduce system administration costs over the lifetime of the system. 

#### 3.4.1. Data Ingestion Throughput Settings

The AWS data pipelines for data ingestion processes in the designed architecture were built upon AWS serverless services such Kinesis Data Streams, Kinesis Firehose, S3 Buckets, AWS Lambda, and DynamoDB. It is crucial for the architecture’s functional requirements to have these services working with the right throughput settings.

The proposed architecture requires data ingestion pipelines that are capable of ingested and persistent data in AWS S3 and AWS DynamoDB 50 requests per second. Each request ingested in this system must have no more than 24 KB (24,576 bytes) payload size in JSON format.
Payload—For each ingestion data pipeline, it is essential to have visibility on the amount of data ingested into the system for a discrete period of time. This depends on the number of incoming requests and the payload has each one of them. The data format used in the system is JSON. Each payload consists of useful metadata and batches of sensor measurement readings. The goal is to have a JSON serialised payload size as close as possible to 24 kb. The prototypes of IoT and Edge devices described in Section 3.2.1 were deployed and tested in a livestock farm located around Troyan city, Bulgaria. During the test period, the collected data were used to create the optimal payload message structure in JSON format (Appendix C). Each livestock IoT device generates around 120 bytes of data for a single measurement. The payload for each smart livestock request from an Edge device consists of 306 bytes of metadata and a batch of 200 IoT single measurements totalling 24,000 bytes. Thus, the total payload size for a single ingestion request is 24,306 bytes, which aligns very well with the system goal.Amazon S3—Using AWS S3 guarantees that the support for thousands of transactions per second upload capacity requirement is achievable. The AWS S3 limitations allow 5500 GET and 3500 PUT/POST operations per second per prefix. However, there is no limit on the number of prefixes in a bucket. Individual S3 objects can be of size from one byte to five terabytes. All this leads to there being no need to take any specific setting actions because the AWS S3 single object size, operating speed, and latency are sufficient for the needs of the proposed architecture to store raw sensor readings and other data for later usage.Amazon Kinesis Data Firehose—There is no need to manage any resources as it is fully managed and provisioned by Amazon. If needed, it can transform data before it is delivered to the destination in the data pipeline. It can scale automatically to satisfy the throughput of the data pipelines using it. The maximum size of a single record is 1024 KB before the record is base64-encoded. As the payload used in the smart livestock system was 24 KB, there was a lot of capacity left. Therefore, there was no need to take any further actions to achieve better throughput.Amazon Kinesis Data Stream—A Kinesis Stream can be scaled to handle from just a few to millions of records per second. Using Amazon Kinesis Producer Library (KPL), the system performs several asynchronous tasks to record data aggregates to increase payload capacity and improve throughput. A single data record pushed to the Kinesis Data Stream is measured in PUT payload units. Every unit is the size of the 25 KB chunk from a data record. In case a record has the size of 70 KB, then it includes three PUT payload units (for 10 KB record size = 1 PUT payload units, etc.). Kinesis Data Streams consists of shards. It is necessary to have enough shards to achieve the required throughput of the designed architecture and to avoid a bottleneck and data pipeline failure. One shard has a maximum ingestion rate of 1 MB/second or 1000 records/second and supports a total data reading rate of 2 MB/second maximum. Every shard was limited by AWS to 5 GET operations/second.

To calculate the volume of data throughputs per second, the formula below is used:VDT = RR × RP(1)
where
VDT—a volume of data throughputs;RR—requirements request per second; andRP—request payload.

The smart livestock architecture requirements are handling 50 requests per second with a request payload of no more than 24 KB per request. Therefore, the amount of data that enters the system is 1200 KB per second. A good practice for robust data pipelines is to have available resources to deal with peak throughputs. Doubling the request per second count results in having a peak bandwidth of 2 × 1200 KB. Therefore, the request size per second is 2400 KB/1024 = 2.34 MB/second. 

The capacity of a stream can be calculated as:TCS = ∑(CSH)(2)
where
TCS—the total capacity of a stream; andCSH—the capacity as shards.

This impacts the shard count as the Kinesis incoming bandwidth is 1 MB/s. Thus, the system needs three shards, therefore TCS = 3 MB.

Amazon DynamoDB—It is always fully managed by AWS. DynamoDB only supports a possible consistency model with a maximum record size of 400 KB, which includes both the binary length of the attribute name (UTF-8 length) and the length of the attribute value.

To determine the initial throughput settings for AWS DynamoDB, the following inputs were considered such as item size, read/write speeds, and read sequence.

AWS DynamoDB supports two types of reads:–Consistent reads where the result cannot reflect all recently completed write operations; and–Strongly consistent reads may have longer latency, thus they may not be available if an outage occurs or there is a network delay.

The unit of recording capacity for storing one record in DynamoDB is not more than 1 KB per second. A unit of strongly consistent read capacity allows the retrieval of one record per second of the database that is not greater than 4 KB. Assume that the record exceeds 4 KB, more than one unit of reading capacity will be needed to read the record from the table. A unit of consistent read capacity enables the retrieval of two records per second from the database where each one is not greater than 4 KB.

The following formulas show how the provisioning of reading/write capacity units are calculated to satisfy the proposed architecture requirements based on the unit size of 4KB and eventually consistent reads:DA = I × RW × 2(3)
where
DA—the amount of data;I—item; andRW—request writes.
WCU = (DA)/WUS(4)
where
WCU—DynamoDB write capacity units;DA—the amount of data; andWUS— units of write capacity.

In the architecture, one request has a payload of 24 KB per item and following the requirements, it was calculated that the total throughput was 50 requests per second with a single request payload size of no more than 24 KB per request. Then, the architecture WCU = (24 × 50 × 2)/1 = 2400 write capacity units.

To calculate eventually consistent reads capacity units, the formula is followed:RCU = (DA)/RUS,(5)
where
RCU—DynamoDB reads capacity units;DA—the amount of data; andRUS—units of reading capacity.

The architecture RCU = (24 × 50 × 2)/4 = 600 eventually consistent reads capacity units, but one unit allows the retrieval of two records per second, therefore RCU = 600/2 = 300 reads capacity units.

Shifting between reading and write capacity models can be performed once every 24 hours. AWS also allows having burst availability beyond the performance provided.

*AWS Lambda function*—As the python code needed for the function execution is less than 3KB and the execution time is less than 100 MS, the AWS Lambda quotas for the amount of available compute and storage resources are sufficient enough to satisfy the architecture requirements.

The proposed settings guarantee throughput capacity for the data ingestion of the measurements of a total 10,000 livestock IoT devices per second calculated as:–50 requests per second;–~24 KB payload per request; and–200 IoT measurements per payload.

#### 3.4.2. Architecture Data Ingestion Rates Tests

A load test was performed to prove that the proposed AWS serverless services settings used in the smart livestock architecture could satisfy the throughput needs for the data ingestion. The scope of the test and the tests strategies are described in Section 2. Materials and Methods—Stage 5.

To cover all strategies, the test was performed using AWS native tools:–AWS Data Generator–Amazon Kinesis Data Stream–Amazon Kinesis Firehose–AWS Lambda–Amazon DynamoDB

##### 3.4.2.1. Test Plan

AWS Data Generator will generate and push data records to Amazon Kinesis Data Stream, which, in turn, will send the data records to a Kinesis Data Firehose delivery stream. Then, the data records will be preserved in Amazon S3. AWS Lambda is subscribed to Amazon Kinesis Data Stream and a function execution is triggered on every push operation to the Kinesis Data Stream. The Lambda in turn pushes the data records to a DynamoDB table. The size and the format of a single data record and the way it is achieved and constructed is explained in Section 3.4.1. Data Ingestion Throughput Settings—Payload.

##### 3.4.2.2. Test Settings and Provisioning

For the test requirements, all resources were created in the Frankfurt (EU-central-1) Amazon availability zone. Amazon S3 Bucket was created with default settings. DynamoDB On-demand table was created with partition key “id” of type string. As the data would be ingested into the system through a Kinesis Data Stream, first the stream was created with three shards and the default data retention period of 24 h (one day). Then a Kinesis Firehose delivery stream was created using as a source the Kinesis Data Stream and as a destination, the S3 Bucket. AWS Lambda function was created, and a trigger was set to the Kinesis Data Stream (Appendix A).

Using the AWS Data Generator requires an AWS Cognito user with a password. For that using cloud formation and a template stored in an S3 Bucket, such a user was created along with all required AWS IAM roles. After successful login into the AWS Data Generator page, the JSON payload sample was used in the record template text box field (Appendix B), region, stream, records per second were set. The Data Generator matches the records per second rate. With all this done, the test was ready to be executed. Steps needed to reproduce the test can be found in Appendix D. 

##### 3.4.2.3. Test Execution and Results

The AWS Data Generator generated and pushed the data to the Amazon Kinesis Data Stream. The test was executed for 30 min starting from 10:15 and ended at 10:45. During the execution of the test, AWS services metrics were monitored in the dedicated AWS console.

Figure 4 shows the total amount of records generated by the AWS Data generator and ingested by the system. Figure 4a shows the average sum (Y-axis) for the data records pushed into the Kinesis Data Stream. The highest amount of data, 397,682,341 bytes, was ingested by the system at 10:25 and the lowest amount of data, 334,072,153 bytes, were ingested by the system at 10:45. Total data of 2,565,165,643 bytes was ingested in the system for the test duration, which makes 28,502 bytes average per second. Figure 4b shows the average latency (Y-axis) measured in milliseconds. The metrics aggregation period defined by AWS is 5 min and is shown on the X-axis on both charts.

Figure 5 shows the Kinesis Data Stream errors. Figure 5a shows the counts (Y-axis) of exceeded throughputs during the reads operations. Figure 5b shows the percentage of failed records (Y-axis) during the puts operations. The metrics aggregation period defined by AWS is 5 min and is shown on the X-axis on both charts.

Figure 6 shows the performance of the Kinesis Data Streams. Figure 6a shows the number of consumed bytes (Y-axis) with a yellow line. The red line shows the maximum number of bytes that can be consumed by Kinesis Data Consumers. Figure 6b shows the iterator age in milliseconds (Y-axis), which tracks the consuming progress of Kinesis Data Stream consumers. Figure 6c,d shows (Y-axis) the average consuming latency in milliseconds and the number of successful consume operations for AWS Kinesis Data Consumers. Figure 6e shows (Y-axis) the count of successful operations per stream. Figure 6f shows (Y-axis) the sum of the incoming bytes with a yellow line. The metrics aggregation period defined by AWS is 5 min and is shown on the X-axis on all Figure 7 charts.

Figure 7 shows the read metrics from Kinesis Data Streams. Figure 7a shows the sum of records read from the stream on the Y-axis. Figure 7b shows in the Y-axis the sum of bytes read from the Kinesis Data Stream. The metrics aggregation period defined by AWS is 5 min and is shown on the X-axis on both charts. 

Figure 8 shows the S3 delivery metrics. Figure 8a shows on the Y-axis the count of throttled messages. Figure 8b shows in the Y-axis the percentage of the delivery success to AWS S3. The metrics aggregation period defined by AWS is 5 min and is shown on the X-axis on both charts.

Figure 9 shows the count of AWS Lambda invocations and the measured performance metrics.

Figure 9a shows the count of AWS Lambda invocations (Y-axis). Figure 9b shows the duration of each invocation (Y-axis). Figure 9c shows along the Y-axis the percentage of successful function invocations with a green line and errors with a red line. Figure 9d shows along the Y-axis the count of throttles during invocations. Figure 9e refers to the percentage (Y-axis) of async innovations success rate. Figure 9f shows the average iterator age during invocations on the Y-axis.

## 4. Discussion

### 4.1. Cloud Architecture

Cloud computing offers computational and storage resources with virtualisation-enabled infrastructure. This allows any application to have high processing power using a large number of processors in the cloud centre and have unlimited storage capabilities. Smart livestock architecture is built entirely in AWS cloud. However, AWS offers more than 200 fully-featured services. There are more than 13 database and storage services, more than 20 management services, and more than 30 data analytics and ML services. The architecture proposed in this research selected these AWS serverless services, which makes it possible to build the required data pipelines with functionalities to ingest a large volume of data from IoT devices and cheaply store unlimited raw sensor and imagery data in AWS S3, where not only are the new data accessible, but also the historical ones. This is of great importance when analytics must be performed again, or ML models need to be retrained and evaluated. AWS DynamoDB storage allows for the fast retrieval of transformed and cleaned data and its subsequent usage for near real-time analytics. The AWS serverless services proposed in the architecture are fully capable of handling future needs in terms of scalability. This result is predictable in terms of cost infrastructure when using the AWS Total Cost of Ownership service, which can perform cost modelling analysis on request. AWS provides tooling for 24/7 monitoring of the deployed services along with automated alerts if certain load thresholds are reached.

Having more data coming from various heterogeneous data sources can make the analytics and predictions of applied machine learning models much more precise and accurate. This is why the extendibility of the architecture data pipelines is another feature that is of excellent value. With it, it is easy to integrate the data from existing IoT systems by doing simple changes in the current data pipeline such as creating a new stream and assigning a unique partition key for the data to be stored in AWS S3.

### 4.2. IoT Devices

Recent research has considered the possibilities of developing cloud and IoT based smart livestock systems because precision livestock farming in agriculture requires sustained production that is not possible by employing traditional systems [47]. The typical architecture of a hardware IoT system consists of sensor and communication modules. The ability of the IoT devices to use WiFi, ZigBee, LoRaWAN, Z-Wave, or other communication protocols makes communication with remote systems and cloud environments possible. In the cases of constrained Internet connectivity, fog or edge computing infrastructure is often proposed as a possible solution for IoT applications in smart farming [48]. However, all these communication protocols are prone to hacker attacks, which can compromise the security of the system.

Current research has taken a substantial step toward addressing the security of IoT devices and Edge computing modules. Unlike many IoT architectures, the smart livestock addresses the IoT devices’ centralised management and security using AWS Greengrass. The service allows full remote device management and the establishment of secure protocols with which every device can be uniquely identified. IoT devices communicate only with the Edge device using LoRa or TCP based protocols, which minimises the risk to be compromised.

The developed prototypes of IoT and Edge devices are also capable of storing the sensor and imagery data if unable to establish a connection. When connectivity is restored, the data are transmitted and there are no losses. The prototypes are also extendable and allow easy integration of additional sensors for required measurements like animal heart rate, gas, and others. 

### 4.3. Architecture Testing

“Even with the most diligent efforts of requirements engineers, designers, and programmers, faults inevitably occur” [49]. This is why tests are so much important, and this is beyond any doubt. Unlike many software architectures and solutions, smart livestock architecture extends this further by proposing a cloud-based architecture implementation that can be tested during the prototype stage, even before the actual implementation and deployment in a production environment. 

The importance of architecture data pipelines and their capability to operate under heavy load is a primary priority. The scope of tests and strategies are described in Section 2. Materials and Methods—Stage 5. The results of the performed test are provided in Section 3.4.2.3. Test execution and results.

The various performance charts in the AWS CloudWatch console provides an excellent insight into the performance of the chosen services. 

Figure 4 shows that the system real-time throughput ingestion rate is quite stable and there are no major fluctuations. As the payload size for the test was 24,306 bytes, the AWS Data Generator was quite close to the requirements and the overall results were credible.

Figure 5 shows quite clearly that during the test execution, there were no failures and the throughput exceeded average value was zero. There was no loss of data records. The system showed that it is capable of handling the required throughput.

Figure 6 shows that data were utilised steadily by the consumers’ streams, which was the expected behaviour. Moreover, the dynamics of Figure 6a and Figure 4a were the same. This means that there was no delay during data consumption by the consumers from streams, which is also proven in Figure 6b, where the value of zero indicates that the records being read are completely caught up with the stream. Looking for spikes or drops in this metric will ensure that the consumers are healthy, and the problems can be caught early. The dynamics of the yellow lines in Figure 6a,f shows the consistency in the performance of the streams.

The consistency in the dynamics in the Figure 7 charts shows the consistency of the payload size of all generated messages used during the test. Figure 7 shows that the number of reads is consistent with the number of bytes read in the dynamics shown in Figure 4a. This consistency is a clear indicator for the data pipeline durability to operate smoothly with the ingested data under heavy loads.

Figure 8 shows the 100% success achieved during the test execution in preserving the ingested data into the S3 Bucket, which aligns with architectural requirements for real-time data storing rates. The count of throttled operations over the test period was always zero, which effectively means that there were no losses of data during high loads, which is the expected behaviour for the architectural requirements.

The charts in Figure 9 show the performance of the serverless functions during the test execution. The average function execution duration was less than 300 MS, which is an excellent speed for partitioning and preserving real-time streaming batches of data into an AWS S3 Bucket and DynamoDB On-demand table. There were also no throttles noticed, the success rate was100% and the error rate was 0%, as expected. 

Although the test execution time was only 30 min, bearing in mind that AWS guarantees the serverless services performance durability and availability as 24/7, the same results can be expected to be achieved regardless of the duration of the tests.

The results clearly showed that the proposed architecture with the provisioned Amazon services is fully capable of handling the required amount of data. The architecture data pipeline was able to persist the ingested data into the AWS S3 Bucket and DynamoDB On-demand table. Both storages operated independently and can be turned off or on without interfering with the work of the other. The total average latency had no fluctuations during the test period and the total count of errors was zero during high loads, which is a clear indicator that the data pipeline is scalable and configured well.

## 5. Conclusions

Despite the current advances in technology, automated smart livestock monitoring systems with minimal human intervention are still considered a struggling phenomenon.

The present study successfully designs architecture for a smart livestock system. All sensitive points of the architecture were successfully tested and the results met the functional requirements of the system. This is a clear indicator that the developed architecture is suitable for further integration in livestock farms.

A key point is that the IoT customers are diverse, and they have quite different requirements based on their use cases. They require very flexible architectures to meet their needs. We would like to achieve complete isolation between hardware components (IoT) and cloud infrastructure. 

The innovative IoT architecture proposed in this article is highly decoupled from the IoT device layer and very flexible in terms of working with different communication protocols, data formats, feature extensibility, etc. This was achieved through the adoption of AWS Greengrass Edge, which extends functionalities by allowing devices to act locally on the data they generate while still taking advantage of the cloud. IoT devices communicate through Edge with the cloud. AWS Greengrass Edge can operate to manage the business logic even if it has lost connection to the cloud and acts like a logical boundary, having a core, devices, and other components that can work together locally. AWS Greengrass core Lambda functions act like glue—they are the coordinators between the IoT devices and the Greengrass core.

However, the Greengrass core has a limitation of working with only 200 IoT devices locally. Nowadays, farms often have more than this number of livestock. It is a tendency for the number of animals on a farm to increase and reach more than 500 animals. By adopting LoRaWan and other communication protocols in the Edge device, this number can easily be handled. Furthermore, local communication is often restricted to a limited distance range. However, livestock can be far away from the farming facilities. LoRaWan can operate with ease over long distances. The LoRa gateway in the Edge device can be in a private LoRaWan network that eliminates the need to filter the messages that are coming from devices outside the system.

Furthermore, when a new system architecture is being designed, not only are the present requirements considered, but also what features may be needed later, probably after a few years. By decoupling the IoT layer with the cloud, extensibility is easily achieved. Regardless of what changes are required in the IoT layer, no matter what new devices and communication protocols there might be after some time, the Edge device can help to adopt the new changes acting as a bridge. 

The architecture also allows for multiple functionalities in a single system so multiple software solutions are no longer needed for different needs. You can now have a security system integrated into smart farming. The latter can be easily extended with ERP or CRM capabilities for every single user if required. Other features and functionalities can be added with ease as the architecture is no longer a boundary.

Finally, to achieve these results in the paper, the following key points can be stated:–A detailed guide for designing smart livestock architecture was developed based on Agile methodology. The essentials are defined in each stage during the design of the architecture such as system requirements, system functionalities, development process following well-designed architecture pillars, tech stack (infrastructure, programming languages, services for the implementation of functional requirements), architecture testing plan (scope, strategies, results in evaluation), and deployment process.–Smart livestock architecture consists of many AWS serverless services that are specially selected and configured in a way to meet the objectives. The relationship between all services in the architecture are visualised in a detailed diagram.–Prototypes of IoT and Edge devices were developed, deployed, and tested. The goal was to test data collection from livestock and to generate the message structure in a JSON format.–The throughput load tests on the developed architecture demonstrated its full capability to handle the required amount of data coming from 10,000 IoT devices per second together with its flexibility and scalability. The results prove the efficiency of the designed architecture and its readiness for implementation in a real environment.

## Figures and Tables

**Figure 1 animals-11-02697-f001:**
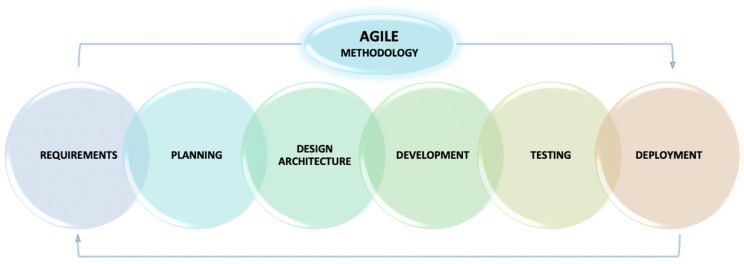
Agile methodology.

**Figure 2 animals-11-02697-f002:**
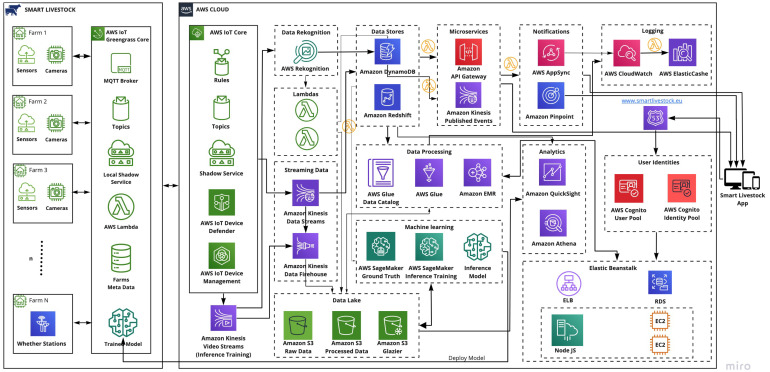
Architecture design for livestock smart farming. (https://www.smartlivestock.eu/, accessed on 20 May 2021).

**Figure 3 animals-11-02697-f003:**
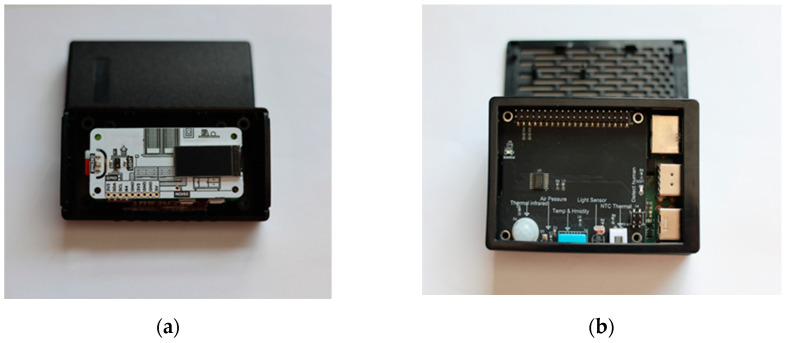
Prototypes of IoT devices. (**a**) Livestock IoT device measures the parameters of livestock and is placed on the cow’s neck. (**b**) IoT Edge device measures the parameters of the environment in a farm.

**Figure 4 animals-11-02697-f004:**
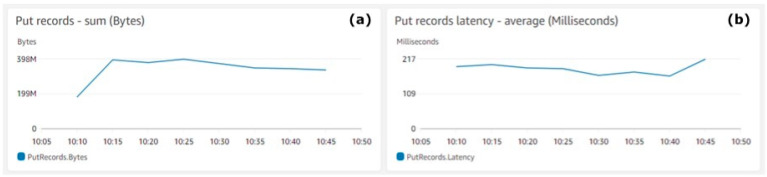
Kinesis Data Stream put records. (**a**) Put records—sum (bytes), (**b**) Put records latency—average (milliseconds).

**Figure 5 animals-11-02697-f005:**
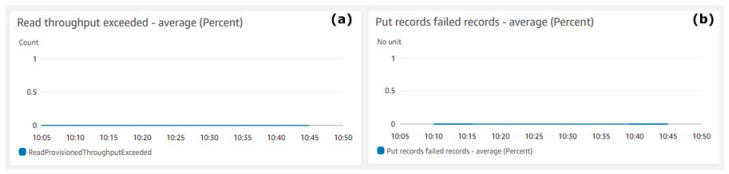
Kinesis Data Stream errors. (**a**) Read throughput exceeded—average, (**b**) Put records failed records—average (percent).

**Figure 6 animals-11-02697-f006:**
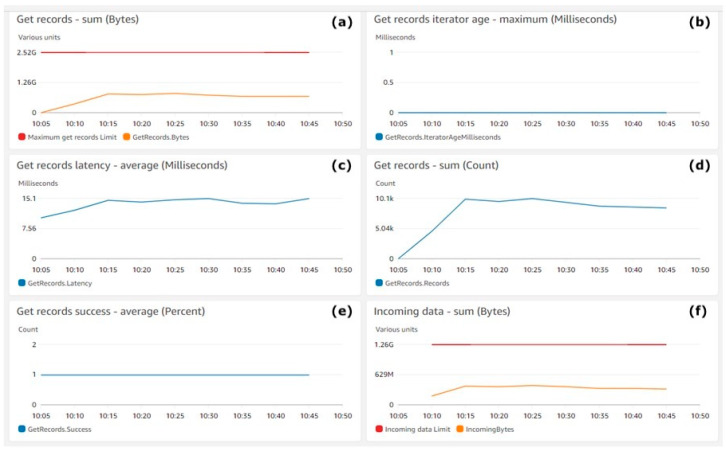
Kinesis Data Stream get records: (**a**) Get records—sum (bytes), (**b**) Get records iterator age—maximum (milliseconds), (**c**) Get records latency—average (milliseconds), (**d**) Get records—sum (count), (**e**) get records success—average (percent), (**f**) Incoming data—sum (bytes).

**Figure 7 animals-11-02697-f007:**
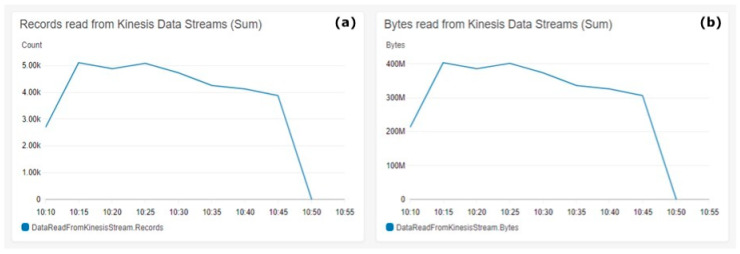
Kinesis Data Firehose reads. (**a**) Records read from the Kinesis Data Streams (sum), (**b**) Bytes read Kinesis Data Streams (sum).

**Figure 8 animals-11-02697-f008:**
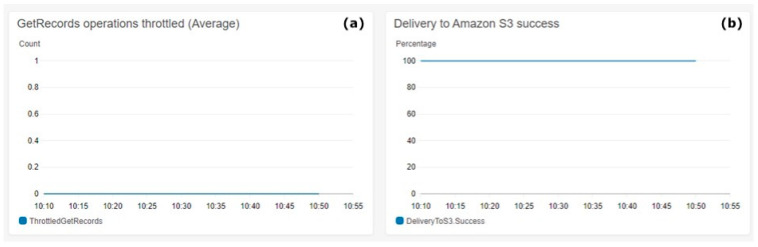
Kinesis Firehose S3 Delivery. (**a**) Get records operations throttled (average), (**b**) delivery to Amazon S3 success.

**Figure 9 animals-11-02697-f009:**
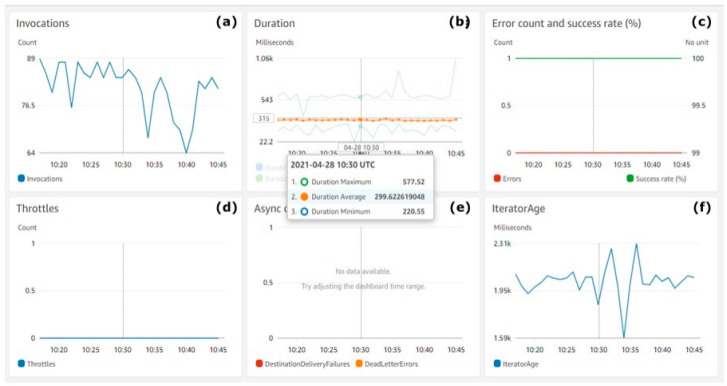
AWS Lambda performance metrics. (**a**) Invocations, (**b**) duration, (**c**) error count and success rate (%), (**d**) throttles, (**e**) async delivery errors, (**f**) iterator age.

**Table 1 animals-11-02697-t001:** Groups of AWS services that fulfil system functional requirements.

Functionalities	AWS Serverless Services
Functionality 1 Ability to integrate and maintain a huge number of IoT devices from both newly developed systems and existing ones	IoT Core Services—a group of services that take care of receiving data and sending user commands to and from IoT devices via the AWS console. Streaming Data Services—services responsible for ingesting data that is generated continuously by thousands of IoT devices. The data are either imagery or represent sensor readings. Data are typically sent simultaneously and in small-sized batches or chunks.
Functionality 2 Ability to store, analyse and process Big Data	Data Lake Services—services allowing the storage of data ingested in the system by IoT devices. Provide the ability to understand what kind of data it is by crawling, cataloguing, and indexing data in the lake. Data Storage Services—services that serve as storage for relational data. Data Processing Services—Services that a responsible for the processing of different types of raw data.
Functionality 3 Ability to model data	Machine learning Services—services working with cleaned data which is used to train machine learning algorithms for regression and classification and obtain ready-trained models.
Functionality 4 Ability to recognition of objects, people, and animals	Data Recognition (AWS)—service working with video and image data and responsible for object recognition.
Functionality 5 Analytical capabilities	Analytic Services—services that are used for data visualisation.
Functionality 7 Ability to record and audit events occurred in the system	Logging Services—services responsible for collecting and analysing events that occurred during the operation of the system.
Functionality 8 Notification capabilities	Notification Services—services that are used for sending notifications to many users at once.

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
