# Peer review of "Design of Scalable IoT Architecture Based on AWS for Smart Livestock"

_animals, 2021, doi:10.3390/ani11092697_

Round 1

Reviewer 1 Report

The paper describes the design of an IoT smart livestock architecture, based on AWS infrastructure, and presents some AWS platform tests. The paper is well structured, it contains very few typos and addresses a very interesting subject.

Paper fails due to the lack of details on the solution (details about sensors, communications), possibly because it is a  in progress work and it has been initiated by the architectural component. There is, in my opinion, a set of questions that should resolve and/or clarify, in order to improve the article.

The architecture raised me some questions:

- Is not clear what live in edge devices and what lives in AWS cloud.

- And there is something that made me think: does MQTT Broker will be outside of AWS? Where did you placed it?  

In the line 316 you refer that Rules system implements some rules for filtering the data that comes from IoT devices. Wouldn’t it to be easier to do it through broker consumer subscriptions? If you instatiate different broker consumers, you could depart data from IoT devices easily, with no extra filtering.

The text refers several technologies but it misses to include references that  would help readers to be aware about what are you writing, and to check each technology details. Please take a look on Word file comments, because It marked all the situations.

In line 346 you state “AWS IoT Device Defender has been chosen in this application because it best fits the needs of the developed application”, but you don0’t justify. And let me tell you I disagree with you,. Why does it best fit. Do you think it is performant in enough? 

There are a few typos highlighted in word document o submitted. Take a look.

In line 522 there is a reference of a noise sensing. Is it audio noise? What do you do(plan to do with)  with such a measure?

In line 533 you show a prototype that senses several different elements, but you do not reference or describe their accuracies. The point is, it is more important to know the transducers ou the sensor accuracies, than seeing the photo.

At line 547 you refer “The communication is of the Device-to-Cloud type. ”, but It does not exist. Computer communications are implemented following a set of well-known standards (named protocols) organized in protocolar stacks, that precisely describe every detail of the communication process, from the bottom layer access technologies (wifi, BLE, ZigBee) up to how the data is encoded (XML, JSON). 

In line 553 you state that “Edge plays a significant role in reducing and transforming the information into the required type and volume, thus significantly reducing the amount of outgoing Internet traffic from the system. “, but you missed to add details about your technical choice. There are several options to perform the communication, and those options have different performances, but you didn’t specify yours.

The edge device illustrated in Figure 1b) is not specified. Additional details are needed here, since the photo just lets us know that it seems to be small. How many Amperes/hour does the appliance consume? How do you power it? How many sensor devices does it supports and connects thoruht the Internet? How does it communicate (3G, zigbee, sattelite, …)? Because this information can give the reader an idea of the use, if there is coverage of communications at the site of a farm, of the difficulty in electrically feeding the device's operation.  

The results of throughput and latency seem to be promising, but I guess reader would prefer testes taking into account Internet connection  generated latency, i.e, the latency time measured was just due to AWS, how is it compared with the full scenario and having Internet connection between AWS and the farm sensors?

You implementation and test description completely ignored Agile methods. You referred agile before, and after presenting the implementation, you recovered the subject. Does it make sense recovering it?

There are a number of issues related to communications that I think you should clear up:

- you miss to specify the communication solution details: how do you transport it, how do you encode it? There are some JSON references, but it is not enough to have a clear idea.

I don’t agree with the way to defend AWS communication solution (lines 874-876) and I think you should change it, because :

-  you don’t need AWS to uniquely identify the sensors. Aren’t you able to draw a solution that uniquely addresses your sensors? There are a lots of radio technologies that offer an MAC address.

- In a IoT solution always exist a gateway (edge accordingly with AWS) that clearly separates the sensor network from the cloud component (and from the Internet).  Not only Amazon allows decoupling, it exist in CoAP, 6TISCH, … 

Author Response

Response to Reviewer 1 Comments

The authors would like to sincerely thank the reviewers for their time to review the submitted manuscript and for the provision of valuable comments.

We did our very best to address them in the revised version of the manuscript.

Revisions in the text related to the reviewers’ comments are in red. 

The authors would like to inform the reviewer that the focus in this manuscript is on the architecture, cloud infrastructure and data pipeline load performance. Edge device and the IoT Livestock devices are in-depth described and explained in another paper that has been approved for publishing but will be printed in the next two months. And this is the reason the paper is not cited in this manuscript. This may also refer to other comments of the reviewer.

Point 1: The architecture raised me some questions:

- Is not clear what live in edge devices and what lives in AWS cloud.

- And there is something that made me think: does MQTT Broker will be outside of AWS? Where did you placed it? 

Response 1:

AWS IoT Core is all about connecting devices to the cloud and its default implementation is based on MQTT protocol and is used to interact with IoT Livestock devices. AWS automatically provide it. However, the Edge device (AWS Greengrass) has a local MQTT broker - Moquette MQTT broker component (aws.greengrass.clientdevices.mqtt.Moquette) which handles MQTT messages between local IoT Livestock devices and the Greengrass core device. As AWS Greengrass 2.0 is built on components, through the MQTT component, MQTT messages are relayed between IoT local devices, local Greengrass publish/subscribe, and AWS IoT Core.

Other components that live on the Greengrass Edge are CLI, StreamManager, Defeneder, Lambdas, Machine Learning components and others.

Add in Manuscript in (Section 3.1.1)

Point 2: In the line 316 you refer that Rules system implements some rules for filtering the data that comes from IoT devices. Wouldn’t it to be easier to do it through broker consumer subscriptions? If you instantiate different broker consumers, you could depart data from IoT devices easily, with no extra filtering.

Response 2:

Data coming from IoT Livestock devices that communicate with the Edge device using LoraWan, cannot use MQTT subscriptions themselves. This data is pushed by the Edge device to a topic “Lora”, where rules trigger specific actions like Lora message decoding, etc.

Add in Manuscript in (Section 3.2.2)

Point 3: The text refers several technologies, but it misses to include references that would help readers to be aware about what are you writing, and to check each technology details. Please take a look on Word file comments, because it marked all the situations.

Response 3:

According to Word file comments, the changes are done.

Point 4: In line 346 you state “AWS IoT Device Defender has been chosen in this application because it best fits the needs of the developed application”, but you don’t justify. And let me tell you I disagree with you. Why does it best fit? Do you think it is performant in enough? 

Response 4:

The authors would like to use services from the AWS ecosystem. Registering IoT Livestock device with AWS IoT Core provides the digital identity of the IoT Livestock Device in the Cloud. The IoT Livestock devices use the certificates to identify themselves to AWS IoT Core. These certificates are generated for it by AWS IoT Core. From that moment on, AWS IoT Device is in charge to maintain and enforce IoT configurations, such as ensuring device identity, authenticating, and authorizing devices, and encrypting device data. “Device Defender sends an alert if there are any gaps in your IoT configuration that might create a security risk, such as identity certificates being shared across multiple devices or a device with a revoked identity certificate trying to connect to AWS IoT Core.” (https://aws.amazon.com/iot-device-defender/)

This is an out-of-the-box solution and the authors do think it is sufficient.

Change in Manuscript:

AWS IoT Device Defender has been chosen in this application because it best fits the needs of the developed application.

Point 5: In line 522 there is a reference of a noise sensing. Is it audio noise? What do you do (plan to do with) with such a measure?

Response 5:

Different parameters reflecting external and internal factors of animal breeding are measured by various sensors. Every phenomenon and technological process that causes stress in animals leads to a change in the hormonal balance in the body, a change in heart rate, blood pressure and a change in the frequency of respiratory movements. The study of the main behavioural activities can provide clarity about the health and physiological comfort of the animals, the presence of stress and disturbance in their biological requirements for technological solutions on the farm. Farm animals are often moved from one group of animals to another. A typical situation is when there is an animal with a weight below the norm; this animal is moved with others in the same condition, where they are treated, for example, with different food, so that these animals reach the weight standards. However, animals build a hierarchy with each other. If there are more dominant animals in the same group, this can often cause stress to other animals and this stress is often expressed in loud sounds. Another example would be if there is not enough food, water or even free space, an animal would express this loudly. A typically calm group of animals keep silent. There are farms where classical music is playing, and the animals enjoy it. When music is over, the animals express this loudly.

Point 6: In line 533 you show a prototype that senses several different elements, but you do not reference or describe their accuracies. The point is, it is more important to know the transducers our the sensor accuracies, than seeing the photo.

Response 6:

The authors use the provided vendors’ documentation which exposes enough data and detailed examples of how to convert sensor readings into scalar values. These readings have been tested and verified under multiple conditions. It is important to state that the bme280 temperature and humidity values reflect the heat emitted from the single-board computer board. In addition, if this happens, the sensor provides a consistently different reading with a large margin from the average readings of similar sensors around it, this sensor is considered faulty, and its readings are not taken into account.

Several sensors are under test currently. Such as

MLX90614ESF GY-906 - contactless IR temperature sensor

Adafruit TMP006 - contactless IR temperature sensor

BerryGPS-IMU V4 – accelerometer and gyroscope

LWIR Thermal Camera for livestock temperature measurement and analysis

MQ-135 gas sensor air quality

DHT22, BME280, and others

Furthermore, during the authors’ stay in Cambridge, multiple discussions has been conducted with Raspberry Pi team members for future intentions to use dedicated, custom-built single-board computer with expanded capabilities which better suit for the system requirements.

Point 7: At line 547 you refer “The communication is of the Device-to-Cloud type. ”, but It does not exist. Computer communications are implemented following a set of well-known standards (named protocols) organized in protocolar stacks, that precisely describe every detail of the communication process, from the bottom layer access technologies (wifi, BLE, ZigBee) up to how the data is encoded (XML, JSON). 

Response 7:

The authors would like to achieve complete isolation between hardware components (IoT) and Cloud infrastructure. IoT Livestock devices communicate only through Greengrass Edge device with the Cloud.

The authors’ intention was to point that they distinguish the following types of communications

  • IoT Livestock device to Edge device using LoRa or TCP based protocols.
  • Edge devise to Cloud using TCP based protocols.

The provided JSON example is for a decoded message for easy understanding of the actual sensor data. LoRa and TCP based protocols encode messages in a different way.

Change in Manuscript:

Device-to-device => Device-to-Edge

Device-to-Cloud => Edge-to-Cloud

Add in Manuscript in (Section 3.2.1)

Point 8: In line 553 you state that “Edge plays a significant role in reducing and transforming the information into the required type and volume, thus significantly reducing the amount of outgoing Internet traffic from the system. “, but you missed to add details about your technical choice. There are several options to perform the communication, and those options have different performances, but you didn’t specify yours.

Response 8:

As it was mentioned in response 1, AWS Greengrass is used to extending functionalities by allowing devices to act locally on the data they generate while still taking advantage of the Cloud. There are two types of IoT Livestock Devices. One type of devices uses WiFi and communicates with the Greengrass Edge by pushing messages to the local MQTT broker. Greengrass Edge device uses MQTT component to transfer data to the AWS IoT Core and back to the IoT Livestock devices. In addition, the authors have included local Lambda functions in the local communication between IoT devices and the Greengrass Edge.

  • IoT devices push messages to a local topic to which a local Lambda is subscribed. This lambda pushes the message to the AWS IoT Core topic.
  • AWS IoT Core pushes a message to an IoT device using the topic. The Local Lambda is subscribed to this topic and gets the message. The local Lambda then pushes the message to a topic to which the IoT device is subscribed.

This approach allows full control over the messages generated by IoT devices. As local Lambdas are deployed remotely, it becomes easy to make changes in the Lambdas logic that controls how IoT messages are handled.

Another type of devices uses LoRaWan and sends messages to a private LoRaWan Network (implemented using ChirpStack) wherein the application server the data is pushed to a topic using MQTT broker.

On the performance side, several use-cases are currently being tested and evaluated:

  • Instead of using ChripStack, AWS LoraWan is using. The advantage is that no infrastructure is maintained, the network server is implemented by AWS, along with many pre-configured Amazon partner devices including gateways, edge devices, development boards, etc. The bad thing is that the concept of an Edge device is lost because the data is received directly in the cloud.
  • Instead of using AWS Greengrass Edge and AWS IoT Core topics, filters, and rules, a custom MQTT broker implementation is made in AWS Cloud. The Edge device is registered in the AWS IoT Core as a regular IoT Livestock device. An MQTT broker is installed that uses the MQTT bridge connection, simply transmits messages from IoT Livestock devices to Cloud and vice versa. Currently, this approach shows the best performance in terms of speed but again the concept of the Edge is lost.

In a conclusion, the authors would like to mention that having the benefits of the Greengrass Edge is considered as important. The capabilities like security, local messaging with local actions, state sync, ML capabilities, security manager, additional stream manager, etc. are a huge plus. However, other approaches to exploring their potential must be given a chance.

Point 9: The edge device illustrated in Figure 1b) is not specified. Additional details are needed here since the photo just lets us know that it seems to be small. How many Amperes/hour does the appliance consume? How do you power it? How many sensor devices does it support and connects through the Internet? How does it communicate (3G, zigbee, sattelite, …)? Because this information can give the reader an idea of the use, if there is coverage of communications at the site of a farm, of the difficulty in electrically feeding the device's operation.  

Response 9:

The edge device is built on top of a single-board computer. Currently, the RaspberryPi4 (shown in the image) has a constant power supply. BeagleBone AI as an Edge device is currently tested. The results will be published when tests are completed. Also, Edge devices from the Amazon Partner devices catalogue are under review. A particularly interesting one is RAK7258 WisGate Edge Lite for the excellent Line-of-Sight LoraWan range specs.

Point 10: The results of throughput and latency seem to be promising, but I guess reader would prefer testes taking into account Internet connection generated latency, i.e., the latency time measured was just due to AWS, how is it compared with the full scenario and having Internet connection between AWS and the farm sensors?

Response 10:

The aim of the authors is to measure the latency in the Cloud infrastructure and in particular to justify that some components of the data pipeline do not act as a bottleneck.

In terms of improving performance, the ongoing tests were mentioned in Response 8.

It is important to note that there is no time-critical use case in the smart farming system. A few milliseconds delay due to network latency is acceptable.

Point 11: Your implementation and test description completely ignored Agile methods. You referred agile before, and after presenting the implementation, you recovered the subject. Does it make sense recovering it?

Response 11:

The intention of the authors is to show how, following the principles of AGILE during the design and implementation of the architecture, the step related to the way the test is performed, documented, and evaluated is indicated. As it is not clear how AGILE was followed, as the reviewer noted, this paragraph has been removed from the manuscript.

Change in Manuscript:

4.1 The first paragraph has been removed

Point 12: There are a number of issues related to communications that I think you should clear up:

- you miss specify the communication solution details: how do you transport it; how do you encode it? There are some JSON references, but it is not enough to have a clear idea.

Response 12:

Response 8 covers that point too.

Point 13: I don’t agree with the way to defend AWS communication solution (lines 874-876) and I think you should change it, because :

-  you don’t need AWS to uniquely identify the sensors. Aren’t you able to draw a solution that uniquely addresses your sensors? There are a lots of radio technologies that offer an MAC address.

- In a IoT solution always exist a gateway (edge accordingly with AWS) that clearly separates the sensor network from the cloud component (and from the Internet).  Not only Amazon allows decoupling, it exist in CoAP, 6TISCH, … 

Response 13:

- Authors’ choice is to use AWS to identify the IoT Livestock devices. Mac address is a possible solution however there are concerns related to security – mac addresses can easily be spoofed. There is no native approach to force the MQTT brokers to filter messages according to the sender’s MAC address. This is how it works. Furthermore X.509 certificate is much safer to be used instead.

- CoAP is primarily a one-to-one protocol. Not yet standardized. It is a Client-Server approach - a client node can command another node by sending a CoAP packet. Although there is support for resource monitoring, CoAP is best suited for a state transfer model, not events.

 If one of the devices is placed behind a firewall, communication between the devices breaks down. Third-party solutions should be used. The authors do not see any benefit in using this approach and going away from the AWS ecosystem.

Kind regards

Reviewer 2 Report

The paper presents the design process of a smart livestock monitoring system architecture.

The authors selected the agile methodology for their development process and the paper describes the different design steps.

The resulting architecture uses almost all the available AWS subsystems without providing any specific innovation. They show how the different amazon services can be used and composed among them.

Reading the paper several curiosities come up about the actual configurations of the different pieces which are not specified

For example, how is Recognition configured? Which events are chosen? How are the different sensor triggers configured?

In summary, I think it is a very good industrial work, where the research part is minimal if not absent at all.

In addition, the paper does not seem to analyse specific livestock farm issues.

I would like the authors to better highlight the innovative contribution to the implementation of the system, showing how the livestock farm problems are solved.
For example, does Rekognition work well? does it depend on livestock density? how is it used? What does Athena analyze? Does it really work to monitor animals? Have they tried it?

Author Response

The authors would like to sincerely thank the reviewers for their time to review the submitted manuscript and for the provision of valuable comments. We did our very best to address them in the revised version of the manuscript. Revisions in the text related to the reviewers’ comments are in red.

Kind regards

Reviewer 3 Report

As it appears from studying the paper, it is interesting in general and addresses to the readership of the journal. In particular the authors introduce the design of an AWS-based architecture for smart livestock monitoring using CPS/IoT equipment and a wide range of cloud services which are detailed described and their tasks according to the application area are clarified. As a proof of concept, a stress test has been performed and the experimental results have shown that the proposed is fully capable of processing the required amount of data architecture with the services provided by Amazon and allow the CPS/IoT infrastructure to use automated scaling mechanisms.

The Introduction of the article provides sufficient background on the subject of research while an adequate number of relevant references is included in it. The system design is clearly described and enhanced by the testing of the system which is enriched with an efficient number of charts. However, for the benefit of the readership the authors should take into consideration the following suggestions:

  1. A paragraph outlining the structure of the article should be added at the end of the introduction section.
  2. Figure 2 should be placed in a separate page in landscape in order to be larger and its elements to be more distinct

Finally, the paper is written in appropriate and understandable English language however some minor spell checking might be needed.

Author Response

The authors would like to sincerely thank the reviewers for their time to review the submitted manuscript and for the provision of valuable comments.

We did our best to address them in the revised version of the manuscript.

Revisions in the text related to the reviewers’ comments are in red. 

Point 1: A paragraph outlining the structure of the article should be added at the end of the introduction section.

Response 1:

Add in Manuscript in Section1

The rest of this paper is structured as follows. Section 2 introduces step-by-step the methodology that is followed and through which key features and requirements are identified to build a well-designed pure Cloud architecture. In Section 3, the proposed AWS Cloud-based architecture with its services and components is described in detail, focusing on scalability, availability, durability, and resilience. The section concludes with the architecture data ingestion pipeline and in-depth throughput load test with a demonstration and explanation of the achieved results. Section 4 presents a detailed discussion about the proposed architecture. Finally, Section 5 concludes the article.

Point 2: Figure 2 should be placed in a separate page in landscape in order to be larger and its elements to be more distinct.

Response 2:

Change in Manuscript: Figure 2 is changed in the landscape position.

Point 3: Finally, the paper is written in appropriate and understandable English language however some minor spell checking might be needed.

Response 3:

Authors have paid a lot of attention to the English language and applied the required stylistic modifications where needed.
